# Home-Based Community Elderly Care Quality Indicators in China: A Systematic Literature Review

**DOI:** 10.3390/healthcare13141637

**Published:** 2025-07-08

**Authors:** Xi Chen, Rahimah Ibrahim, Yok Fee Lee, Tengku Aizan Hamid, Sen Tyng Chai

**Affiliations:** 1Malaysian Research Institute on Ageing, Universiti Putra Malaysia, Serdang 43400, Malaysia; gs57714@student.upm.edu.my (X.C.); chai@upm.edu.my (S.T.C.); 2Department of Human Development and Family Studies, Faculty of Human Ecology, Universiti Putra Malaysia, Serdang 43400, Malaysia; 3Department of Government and Civilization Studies, Faculty of Human Ecology, Universiti Putra Malaysia, Serdang 43400, Malaysia; leeyokfee@upm.edu.my; 4Academy of Science Malaysia, Kuala Lumpur 50480, Malaysia; tengkuaizan06@gmail.com

**Keywords:** community care, elderly care, quality indicators, service evaluation, China

## Abstract

**Background**: China’s rapidly aging population has increased the need for effective community-based eldercare services. However, the lack of standardized, culturally relevant evaluation frameworks hinders consistent service quality assessment and improvement. **Objective**: This systematic review aims to identify, synthesize, and critically evaluate the existing quality indicators (QIs) currently utilized for home-based community elderly care HCEC in China. It also aims to highlight gaps to inform the development of a more comprehensive and context-appropriate quality framework. **Methods**: Following PRISMA guidelines, systematic searches were conducted across Web of Science, PubMed, Wiley, and CNKI databases for studies published in English and Chinese from 2008 onward. Extracted QIs from eligible studies were categorized using Donabedian’s structure–process–outcome (SPO) model. **Results**: Fifteen studies met the inclusion criteria, with QI sets ranging from 5 to 64 indicators. Most studies emphasized structural and procedural aspects, while outcome measures were limited. Key gaps include inconsistent terminology, insufficient medical care integration, narrow stakeholder engagement, and limited cultural adaptation of Western theoretical frameworks. Furthermore, subjective weighting methods predominated, impacting indicator reliability. **Conclusions**: Currently, there is no formal quality framework to guide service providers in HCEC, and therefore, quality indicators can be described as fragmented and lack cultural specificity, medical integration, and methodological robustness. Future research should prioritize developing culturally anchored and medically comprehensive QI frameworks, standardize indicator terminology, actively involve diverse stakeholders through participatory methods, and adopt hybrid methodological approaches combining subjective expert insights and objective, data-driven techniques. Alignment with established international standards, such as the OECD long-term care quality indicators, is essential to enhance eldercare quality and support evidence-based policymaking.

## 1. Introduction

The phenomenon of aging is a significant global challenge driven by increased longevity and declining fertility rates [1], positioning the care and health of older adults as crucial international issues. China, with the largest and fastest-growing elderly population globally, faces particularly acute challenges. By 2024, individuals aged over 60 comprised about 22% (310 million) of China’s population, while those over 65 accounted for approximately 15.6% (220 million), highlighting the urgency of effective eldercare strategies (National Bureau of Statistics of China 2025) [2].

Community-based eldercare has emerged globally as a preferred approach due to its alignment with older adults’ preferences for aging in familiar home environments [3]. However, China’s one-child policy, implemented from 1980 to 2015, has significantly diminished family sizes and, to a certain extent, altered traditional family structures, increasing caregiving pressures on family members who traditionally shoulder eldercare responsibilities due to the rooted tradition of filial piety. This demographic shift, coupled with the high prevalence of chronic conditions and disabilities among the elderly population (Su et al., 2023; Chen et al., 2024; Wei et al., 2019) [4,5,6], presents considerable caregiving challenges, as nearly 100 million older adults live alone or in “empty nest” situations (Peng 2017) [7].

In response, an evolving multi-tiered eldercare model integrating home-based services, community support, and medical-nursing services has emerged. Home-based community eldercare in China refers to a hybrid care model designed to enable older adults to age in place, incorporating in-home services (e.g., personal care and health monitoring) and community-based supports (e.g., daycare, rehabilitation, and social activities) to maintain independence and reduce institutional reliance (He & Wei 2023; Ying et al., 2024; Asian Development Bank 2023) [8,9,10]. In China, HCEC services are delivered through a collaborative framework involving government agencies, community organizations, healthcare providers, and private enterprises. As these services can be accessed through different service providers with different characteristics (private/public, big/small, and geographical locations), the quality and standard of care given are expected to vary, highlighting the need for systematic evaluation. The existing literature identifies several critical gaps in evaluating community eldercare quality, including limited satisfaction rates among elderly individuals (Yao 2021) [11], insufficient management skills (Qing 2016) [12], and inadequate professional training (Wen 2013) [13]. Research on quality indicators (QIs) for community eldercare primarily emphasizes the structural and procedural aspects (Liu 2020) [14], with limited comprehensive evaluation of integrated medical and healthcare services. Despite the development of home-based community care services, comprehensive evaluation frameworks specifically tailored to assess the quality of these services remain limited.

Quality indicators (QIs) play an essential role in evaluating and improving eldercare services. Internationally, organizations such as the Organisation for Economic Co-operation and Development (OECD) have established comprehensive long-term care indicators covering domains like safety, effectiveness, person-centeredness, coordination, and equity of care [15,16,17]. In contrast, the prevailing Chinese frameworks are predominantly adopted from Western models, such as SERVQUAL and Penchansky & Thomas’s 5A model, but rarely systematically define and utilize specific measurable indicators tailored to China’s unique cultural and societal contexts. Moreover, research efforts in China have predominantly emphasized theoretical development, leaving the systematic identification and critical evaluation of practical QIs underexplored. Therefore, the main objective of this study is to systematically identify, synthesize, and critically evaluate the specific quality indicators currently employed in home-based community elderly care in China. This systematic review addresses the central research question:

What specific quality indicators have been used in evaluating home-based community elderly care in China, and how can these indicators inform the development of a comprehensive, culturally relevant, and practically applicable quality assessment framework aligned with international standards?

By explicitly analyzing the indicators themselves, rather than primarily focusing on theoretical models, this review seeks to clarify current evaluation practices, identify critical gaps, and provide evidence-based insights to support future policy and practice enhancements in eldercare quality assessment. This paper is the first systematic review explicitly examining quality indicators for community-based elderly care in China, thus providing valuable insights for improving eldercare service delivery.

## 2. Materials and Methods

### 2.1. Search Strategy

The systematic search strategy was clearly structured in March 2025 using the PICO framework to ensure clarity and transparency:

Population (P): Elderly adults aged 60 years and older.

Intervention (I): Quality indicators (QIs) used to evaluate home-based community elderly care.

Comparison (C): Not applicable.

Outcomes (O): Identification, synthesis, and critical evaluation of relevant QIs.

Systematic searches were conducted across four major databases: Web of Science, PubMed, Wiley, and the China National Knowledge Infrastructure (CNKI). Detailed search strategies used for each database were as follows:

Web of Science: (“community care” OR “home care”) AND (“elderly” OR “aged” OR “older adults”) AND (“quality indicators” OR “evaluation” OR “quality criteria”)

PubMed: ((“community care”[Title/Abstract] OR “home care”[Title/Abstract]) AND (“elderly”[Title/Abstract] OR “aged”[Title/Abstract]) AND (“quality indicators” OR “evaluation”))

Wiley: (“community-based elderly care” OR “home-based elderly care”) AND (“quality indicators” OR “service quality evaluation” OR “evaluation criteria”)

CNKI (Chinese language): (“home-based community eldercare” OR “home-based eldercare services” OR “community eldercare services”) AND (“quality indicators” OR “evaluation of service quality” OR “assessment of service quality”)

Furthermore, manual searches (snowballing) were performed through reference lists of the included studies to identify additional relevant literature.

### 2.2. Study Selection

Peer-reviewed studies published from 2008 onwards were included, aligning with significant policy developments in China regarding community elderly care services.

Inclusion criteria:Articles published in English or Chinese to capture local and international insights.Focused specifically on populations aged 60 years and older.Studies developing and implementing quantitative tools or instruments for measuring quality indicators of community eldercare in mainland China.

Exclusion criteria:

Articles focusing on institutional care, nursing homes, or exclusively addressing cognitively impaired or palliative care populations, as these involve distinct and specialized care requirements not generalizable to broader community settings.

The systematic literature review was structured in four main stages according to the PRISMA guidelines (see Figure 1):Identification: Initial database searches yielded 4081 articles. An additional 12 articles were identified through reference list checks (snowballing technique). After removing duplicates, 2556 articles remained for further screening.Screening: Titles and abstracts of the 2556 articles were screened by one researcher. Of these, 2502 articles were excluded due to irrelevance or failure to meet the initial inclusion criteria.Eligibility: Full-text screening was conducted by two independent reviewers on the remaining 54 articles (33 Chinese, 21 English). Articles were excluded due to irrelevant topics, exclusive QI development without empirical testing, focus on non-community- or non-home-based care, non-exclusive focus on older people, not applicable to mainland China, satisfaction studies unrelated to quality indicators, and inaccessible full texts.Included: Fifteen quantitative studies met the inclusion criteria. Inter-reviewer agreement at the full-text inclusion stage was very high, achieving a kappa statistic of 0.83 [18].

### 2.3. Data Extraction and Analysis

Articles were screened initially by title and abstract by one researcher; two researchers independently conducted full-text reviews. Data management and analysis were guided by PRISMA guidelines. All identified articles underwent a structured data extraction process using standardized forms, capturing author information, publication year, geographic location, study aims, sample sizes, quality indicator systems developed, validation methods, theoretical models, and weighting techniques. Data extraction accuracy was independently verified by two reviewers, and discrepancies were resolved through consensus or arbitration by a third reviewer.

Moreover, extracted quality indicators were categorized according to Donabedian’s framework—structure, process, and outcome—providing a structured analysis approach. This allowed a clear comparison of indicators and identification of key gaps, particularly regarding medical care integration and stakeholder diversity in evaluation methods.

### 2.4. Quality Assessment

A quality assessment was performed on the 15 included studies using PRISMA guidelines, focusing on clarity of objectives, methodological rigor, validity and reliability, appropriateness of statistical analyses, result reporting, and robustness of conclusions. The methodological rigor varied significantly; while approximately half demonstrated strong methodological approaches, others faced limitations such as small sample sizes and inadequate validation processes. Around 60% explicitly reported validity and reliability through expert reviews and theoretical frameworks. Most studies appropriately employed statistical methods, though some lacked detailed explanations. Result reporting was generally comprehensive, though inconsistencies in terminology were noted.

### 2.5. Protocol and Registration

This systematic review was not registered with any formal protocol database such as PROSPERO. The review was conducted independently in accordance with PRISMA 2020 guidelines to ensure methodological transparency and rigor.

## 3. Results

### 3.1. Geographical Distribution

Geographical distribution varied significantly, with the majority conducted in Shaanxi Province (39.1%), followed by Beijing (13%); Jiangsu Province (13%); Hebei Province (8.7%); Tianjin (8.7%); and one study in each of these regions in Jiangxi, Jilin, Guangzhou, and Shanghai (Figure 2).

### 3.2. Description of the Studies

Data from the included studies are summarized in Table 1, which clearly lists only the authors, year of publication, and unique identifiers assigned to each study. Comprehensive details, including study titles, locations, aims, sample sizes, indicator systems developed, total number and categorization of quality indicators (QIs), validation methods, theoretical models, and weighting methods, have been relocated to Appendix A for enhanced readability and ease of cross-referencing [19,20].

### 3.3. Characteristics of the Quality Indicators

Indicators were categorized based on Donabedian’s model into structure, process, and outcomes [36], comprising 29% structure, 47% process, and 24% outcome indicators. Table 2 outlines specific indicators within each category, highlighting the diverse approaches used and emphasizing significant gaps, particularly in integrated medical care assessments [14].

## 4. Discussion

This systematic review analyzed 15 studies on quality indicators (QIs) for home-based community elderly care (HCEC) in China. The findings identified core patterns and gaps that inform future development of comprehensive evaluation frameworks tailored to the Chinese context.

### 4.1. Theoretical Foundations and Cultural Relevance

A well-grounded theoretical framework is essential for designing effective QIs. However, only two-thirds of the studies reviewed explicitly referenced theoretical models such as SERVQUAL [27,28,30], the 5A model [22,29,32], 3C theory [37], and key performance indicators (KPIs) [34]. Zhu (2019) and others advanced this by integrating structure, process, and outcome dimensions via the SPO model [35]. Despite this progress, overreliance on Western theories raises concerns about contextual and cultural applicability. Without integrating Chinese cultural values and industry-specific conditions, indicator systems risk being misaligned with local care practices (Li 2015) [26]. Traditional Chinese cultural concepts such as filial piety, Confucian ethics, and community mutual aid significantly influence eldercare practices in China. For example, filial piety emphasizes children’s duty to care for elderly parents, and Confucian ethics stress respect, moral obligation, and social harmony. Operationalizing these values could involve QIs focused explicitly on family engagement (frequency of family visits and satisfaction of elderly individuals with family participation), mutual community support (community-organized social care activities and community involvement in caregiving), and indicators evaluating respectful, dignity-focused interactions between care providers and elderly individuals.

Future indicator development should therefore integrate these culturally grounded elements to create meaningful and relevant evaluation frameworks that genuinely reflect the lived experiences and societal expectations of elderly individuals in China.

### 4.2. Integration with Healthcare Services

Many studies emphasized environmental and service aspects of care, yet few addressed medical care integration. Only two studies evaluated healthcare and nursing services within HCEC, an evident shortfall given the high prevalence of chronic diseases among older adults [23,35]. Current indicators largely overlook comprehensive medical and nursing care.

To bridge this gap, it is essential to develop indicator systems explicitly incorporating medical care metrics, such as chronic disease management, emergency response effectiveness, and integrated care coordination. Actionable recommendations include creating collaborative platforms involving healthcare providers and community eldercare services to ensure holistic service delivery.

Furthermore, the terminology across domestic studies lacks consistency, creating barriers to effective assessment (Liu 2017) [38]. Moving forward, indicator systems should incorporate standardized definitions and professional medical metrics to ensure precision and comprehensiveness.

### 4.3. Methodological Approaches to Indicator Weighting

The review revealed a predominant use of the analytic hierarchy process (AHP) for indicator weighting [21,22,23,29,31]. While AHP provides a structured approach, it remains largely subjective. A limited number of studies explored complementary methods such as Delphi and expert consensus meetings, but reliance on traditional expert panels risks perpetuating bias (Guo 2014) [39].

To improve methodological rigor, future research should explore hybrid models that combine subjective (expert-based) and objective (data-driven) methods. Techniques such as compromise coefficient weighting, entropy method, and the Frank–Wolfe algorithm may help assign weights more reliably, improving the precision and credibility of resulting evaluations (Han 2010) [40].

### 4.4. Scope and Application of Indicator Systems

Most reviewed studies were geographically limited to specific provinces or municipalities, restricting the generalizability of their findings [24,25,33]. Furthermore, quality assessments predominantly emphasized the elderly individual’s perspective, largely neglecting critical insights from caregivers, family members, healthcare providers, and service administrators. Such limited perspectives risk overlooking the multifaceted and context-sensitive nature of eldercare quality, given that subjective perceptions vary significantly based on personal experiences and health status (Wu 2018) [41]. Moreover, reliance on predominantly quantitative designs further constrains nuanced interpretations (Tang 2018) [42]. To address this significant gap, future research should employ participatory quality indicator (QI) design methods. Specifically, stakeholder workshops; Delphi panels; or focus group discussions involving caregivers, healthcare providers, elderly individuals, and administrators can facilitate the co-creation of indicators, ensuring diverse perspectives and practical considerations are effectively integrated. Additionally, qualitative methods such as in-depth interviews or case studies should be systematically incorporated to capture underlying barriers and challenges encountered by these stakeholders, such as caregiver burden, resource constraints, and service delivery complexities (Liu 2020) [14]. Incorporating community-level risk-adjustment factors will further enhance the contextual accuracy and applicability of quality assessments (Guo 2014) [43].

### 4.5. Comparative Frameworks and International Standards

Unlike established international frameworks that emphasize comprehensive, integrated health and social care assessments (e.g., the OECD’s long-term care quality indicators), there is a lack of systematic evaluation indicators to guide service providers, resulting in the absence of standardized frameworks governing community-based eldercare. The current practice adopted by service providers often narrowly focuses on structural or procedural aspects without robustly integrating medical and caregiving components. The OECD’s (Organization for Economic Co-operation and Development) long-term care quality indicators are a set of internationally recognized metrics can be adapted for the community-based care services in China, since these indicators are part of the OECD’s broader initiative to improve aging and eldercare policy by promoting transparency, accountability, and evidence-based decision-making [15,16,17]. Table 3 outlines the indicators comparison between OECD and the current practice in China. This comparison provides a clear understanding of the current state of community-based care services in China and the standards of OECD, thus paving the way forward for the development of the community-based elderly care in China.

The comparative analysis reveals critical gaps and alignments between Chinese HCEC practices and OECD benchmarks. For instance, in the domain of safety, OECD indicators focus explicitly on standardized tracking of falls, pressure ulcers, and avoidable hospitalizations. Chinese practices show limited and inconsistent adoption of such systematic safety tracking. Regarding person-centered care, OECD emphasizes respect for individual preferences and comprehensive care planning, whereas Chinese indicators primarily address general satisfaction without clear operational measures of personalized care planning. Similarly, OECD effectiveness indicators prioritize the systematic tracking of functional status improvement and pain management, areas minimally addressed in current Chinese frameworks.

To align Chinese eldercare practices more closely with the OECD standards, future QI frameworks should explicitly adopt indicators for systematic safety and health outcome tracking, personalized care planning, and structured care coordination.

## 5. Conclusions

This systematic review analyzed 15 studies on quality indicators (QIs) for home-based community elderly care (HCEC) in China, identifying critical patterns and gaps. The findings revealed that, while the existing frameworks have made progress by adopting models such as SERVQUAL, SPO, and KPIs, there remains an overreliance on Western theories with limited integration of Chinese cultural norms and familial expectations. Medical care integration within HCEC indicators is notably insufficient, and methodological approaches are predominantly subjective, relying heavily on expert opinion through the AHP and Delphi methods. Additionally, the scope of the existing studies is geographically narrow, lacks diverse stakeholder perspectives, and demonstrates limited methodological diversity. Comparisons with OECD frameworks further highlight the gaps in safety, effectiveness, person-centeredness, care coordination, and equity monitoring within China’s HCEC indicators.

To effectively address these gaps and enhance eldercare quality, future research and practice should:Develop culturally grounded theoretical models that integrate local norms with internationally recognized best practices, ensuring relevance and global alignment.Expand and standardize indicator systems to include medical care components, safety tracking (e.g., fall rates), health outcomes, individual preferences, autonomy, care pathway integration, equity, and access.Utilize advanced methodological approaches combining subjective expert insights and objective data-driven strategies to improve the reliability and validity of evaluations.Include diverse stakeholder perspectives—elderly individuals, caregivers, healthcare providers, and management—and adopt mixed methods approaches for a more comprehensive and nuanced understanding of eldercare quality.Promote comparative evaluations both horizontally (across regions) and vertically (across service tiers), incorporating risk adjustment mechanisms to account for demographic and regional variations.Alignment with International Standards: Integrate elements from established international frameworks, such as the OECD’s long-term care quality indicators, to enhance comprehensiveness and facilitate global benchmarking.

Despite its valuable insights, this review has certain limitations. First, it focuses solely on peer-reviewed literature, potentially overlooking relevant policy reports. Second, the scope is limited to studies published in Chinese and English, which may exclude pertinent research in other languages. Third, the heterogeneity of the study designs and quality among the included articles poses challenges for direct comparison. Lastly, the review emphasizes indicator frameworks without analyzing real-world implementation outcomes, limiting practical insights.

Nonetheless, the insights provided by this systematic review highlight critical opportunities for future research and policy development aimed at improving the quality and effectiveness of home-based community elderly care in China, ultimately enhancing the quality of life for the nation’s rapidly expanding elderly population.

## Figures and Tables

**Figure 1 healthcare-13-01637-f001:**
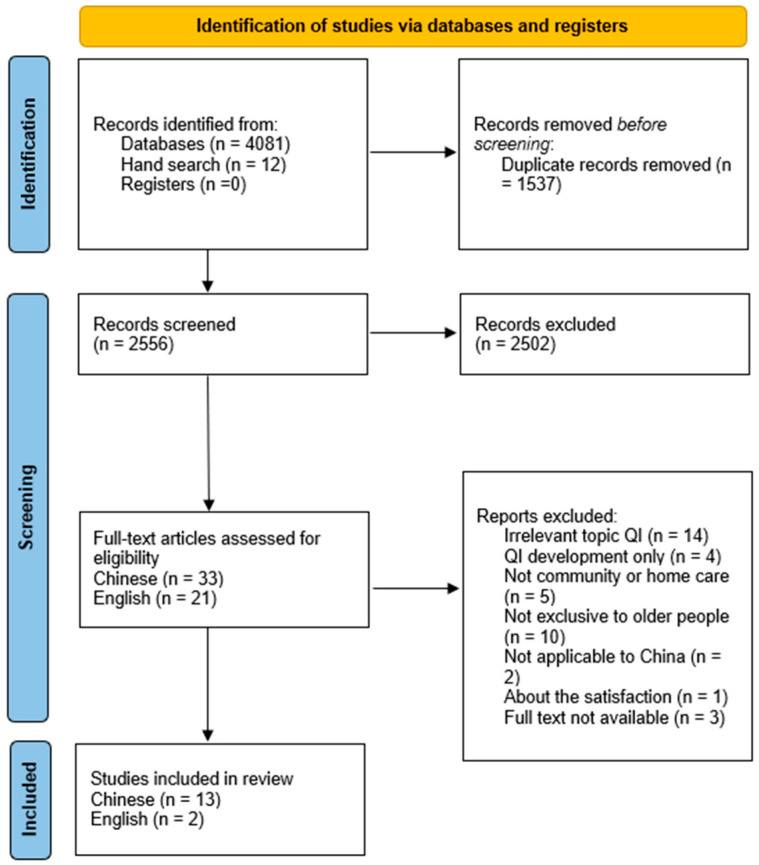
PRISMA flow diagram of the study selection, inclusion, and exclusion.

**Figure 2 healthcare-13-01637-f002:**
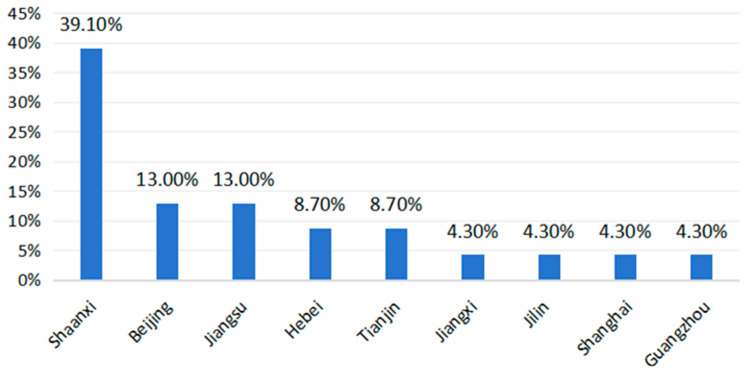
Geographical distribution of the identified studies.

**Table 1 healthcare-13-01637-t001:** Summary of the included studies.

ID	Author	Year
1	Xiaoqin Li	2016 [21]
2	Jinrong Hu	2022 [22]
3	Luo Yu	2018 [23]
4	Cai Zhonghua	2016 [24]
5	Li Congrong	2019 [25]
6	Hao Li	2015 [26]
7	Song Fengxuan	2014 [27]
8	Wen Haihong	2019 [28]
9	Yong Lan	2018 [29]
10	Gan Lu	2023 [30]
11	Zhang Xiaoyi	2012 [31]
12	Ma Duoduo	2023 [32]
13	Xu Qiang	2019 [33]
14	Wen Haihong	2017 [34]
15	Zhu Liang	2019 [35]

**Table 2 healthcare-13-01637-t002:** Characteristics of home-based community elderly care QIs across the identified QI sets.

Donabedian Model	Indicator Type	Quality Indicators	Articles ^1^
Structure	Organizational characteristics	∙ Financial resources (government funding, social assistance, and the proportion of general administrative expenses in total expenditure) ∙ Operating management of users (reasonable charging standards, personalized services, appointment system, formal service contracts and procedures, recording information of elderly individuals, complaint rate, implementation of preventive measures, emergency plan, implementation of government’s mandatory tasks, reports of accidents, supervise and feedback and evaluation system, information service system and database, WeChat platform, reputation, and visibility) ∙ Management of staff (receiving regular national inspection, providing staff training, clear service guidelines, service supervision and inspection mechanism, employee reward and punishment incentive mechanism, clear responsibilities of personnel, and complete rules and regulations) ∙ Older people (the elderly have the right to make decisions and participate in activities decision-making, the proportion of the elderly with pension)	5, 9, 14, 15
	Infrastructure equipment	∙ Basic facilities and equipment (walk assistant equipment, clean assistant, pager, safety protection measures and auxiliary use of notes, barrier free space, first aid facilities and equipment, numbers of beds, basic medical equipment, and observation room) ∙ Advanced facilities and equipment (spiritual comfort, fall prevention of corridors and stairs, emergency call system, information equipment, and smart bracelet) ∙ Timely inspection and maintenance ∙ The atmosphere of humanistic care	2, 3, 5, 6, 8, 9, 10, 12, 13, 15
	Environmental sanitation	∙ The indoor environment (clean and comfortable, light and temperature and humidity are appropriate, the disinfection frequency, room layout is reasonable, and privacy) ∙ The surrounding environment (safe and quiet, the greening rate, the disinfection frequency, and accessibility of emergency traffic services)	3, 7, 14, 15
	Staff	∙ Personal information (numbers, education background, welfare, contract of labor, turnover rate, satisfaction, and performance appraisal system) ∙ Work professionalism (professional certificate, professional knowledge and skills, professional dedication, warm respect for elderly individuals, trustworthy and patient and reliable, clean and tidy clothes, understand the needs of the elderly and their families, attach importance to the difficulties encountered by the elderly and are willing to help, provide services timely and actively and accurately, complete the promised services on time, safeguard the rights and dignity of elderly individuals, standardize the service process of the staff, and volunteer plan) ∙ Professional skills (psychological consultation and law assistant consultation)	2, 5, 6, 7, 9, 10, 12, 14, 15
	Community qualifications	∙ Administrative license ∙ Food business license ∙ Medical qualification	2, 14
Process	Day care and home care service	∙ Day care (healthy and timely meal service from online and offline, walk assistant, laundry services, haircut and trim nails, transport assistant) ∙ Home care (laundry services, indoor cleaning, walk assistant, laundry services, haircut and trim the nails, wear clothes, assist toilet, emergency service, bath assistant service, number of baths per month, hotline service, full-time housekeeper service, elevator service, hourly worker service, milk delivery service, home care bed service, home appliance maintenance service, liquefied gas replacement services, and drains dredging services)	1, 2, 4, 5, 7, 10, 13, 14, 15
	Entertainment services	∙ Space (entertainment places, fitness facilities, outdoor area for activities) ∙ Outdoor activities (gentle sports, sight-seeing and tour) ∙ Cultural activities (reading, drawing, calligraphy, internet, cards, and handwork) ∙ Entertainment activities for the disabled elderly ∙ Records of entertainment activities	1, 3, 4, 7, 13, 14, 15
	Spiritual comfort	∙ Communicate (company with the elderly and communicate with elderly individuals) ∙ Psychological (provide psychological counseling, evaluation of psychological problems of elderly individuals, formulation of psychological crisis intervention plan, degree of specialization of psychological counseling, and the number of psychological health lectures) ∙ Emotion (provide spiritual comfort services, emotional counseling, social meeting for single senior citizens services, hospice care services, and respect the cultural customs of the elderly and their families) ∙ Problems (incidence of insomnia, incidence of anorexia and hunger strike, incidence of self-abuse and suicide, and incidence of drug refusal)	1, 4, 5, 7, 13, 14, 15
	Elderly healthcare services	∙ Nursing services (daily professional nursing services, acute disease nursing services, qualified rate of basic nursing, qualified rate of primary nursing, qualified rate of nursing staff with certificate, nursing station, nursing evaluation, grading nursing standards and grading nursing service specifications, qualified rate of nursing operative technique, correctly carry out relevant medical and nursing activities, rational use of drugs and home safety guidance services, assist the elderly to properly keep and use drugs according to the doctor’s advice, infection control, correct emergency treatment for common accidents or emergency, correctly assist the elderly to move their body position, correctly use wheelchairs and other walking aids, and medication guidance for chronic diseases) ∙ Health status (appearance, smell, clothes, mood, consciousness, weight monitoring, blood glucose monitoring, blood pressure monitoring, health risk assessment, reevaluation when the function changes, incidence rate of falls per year, incidence rate of other accidents per year, accurate record of physical condition, health record filing, and others in the minimum data set)	1, 15
	Combination of medical care	∙ Basic treatment (outpatient treatment, drug injection, oral hygiene, dispensing medications, physical examination, patient compliance instruction, clear medical diagnosis, correct and effective treatment and nursing, health consultation, and basic emergency treatment) ∙ Further treatment (exclusive medical services, hierarchical diagnosis and treatment, timely contact and referral, comprehensive medical programs, whether it can diagnose and treat the severe elderly, whether it has built-in geriatric hospital, rehabilitation hospital, nursing home, and traditional Chinese medicine hospital or hospice care institution) ∙ Medical insurance (whether the built-in medical institution is included in the basic medical insurance and the proportion of the elderly with medical insurance) ∙ System of medical care (the convenience to apply for and obtain medical and nursing services, management and treatment and observation and record of chronic diseases, number of diseases that can provide medical services, two-week hospital visit rate, and whether the proportion of professional personnel is above 85%)	1, 3, 4, 5, 6, 7, 13, 15
	Rehabilitation healthcare	∙ Facilities (the number of rehabilitation equipment, and rehabilitation healthcare room) ∙ Rehabilitation implementation (the elderly rehabilitation training (massage, exercise for power, physiotherapy), evaluate the elderly dysfunction, elderly rehabilitation guidance, postoperative rehabilitation nursing service, and rehabilitation activity times monthly)	1, 3, 5, 13, 15
	Education services	∙ Health lectures (nutrition, regimen, prevention of common diseases, chronic disease, rehabilitation knowledge, and family care) ∙ Other lectures (law assistant, safety knowledge, and death education) ∙ Health education plans and implementation records, and the number of health education activities per year	1, 4, 6, 14, 15
	Accessibility	∙ Distance and cost (community service center, elderly law assistant center, activity center, and elderly school) ∙ Convenience (community care and home care) ∙ Waiting time of appointment	2, 9, 12
Outcome	Satisfaction	∙ Satisfaction of environment (basic equipment, safety, public space, housing, and the planting of public area) ∙ Satisfaction of service (day care, healthcare, safeguard, cultural and sports, spiritual comfort, law assistant, charity, and staff and volunteer training) ∙ Satisfaction of emotion (community elderly culture, a sense of community, and social participation) ∙ Satisfaction of rights and interests protection ∙ Satisfaction of the elderly family members	2, 3, 6, 12, 13, 14, 15
	Affordability	∙ Day care (meal and walk assistant) ∙ Home care (meal, cleaning, and walk assistant) ∙ Healthcare ∙ Spiritual comfort ∙ Elderly association and elderly school	2, 10, 12
	Assessment	∙ Completion rate ∙ Record ∙ Quality assessment and improvement	3, 15
	Expectation	∙ Community service ∙ Community construction ∙ Community environment	14

^1^ Article ID numbers from Table 1. QIs: quality indicators.

**Table 3 healthcare-13-01637-t003:** OECD vs, the current practice in China.

OECD Indicator Domain	Typical OECD Indicators	Current Practice
Safety	Falls, pressure ulcers, avoidable hospitalizations	Limited focus; some mention of emergency plans, accident reports
Effectiveness	Pain management, functional status improvement	Some inclusion of health monitoring (e.g., BP and glucose); lacks systematic outcome tracking
Person-Centeredness	Satisfaction, respect for preferences, care planning	Satisfaction indicators present; few indicators on personalization or care plans
Coordination of Care	Health–social care integration, continuity of care	Rarely assessed; minimal integration of hospital and community data
Access and Equity	Timeliness, regional/economic disparities	Not systematically addressed; limited evidence of equity monitoring

OECD: Organization for Economic Co-operation and Development; BP: blood pressure.

## Data Availability

No new data were created by this study.

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
