# Peer review of "Home-Based Community Elderly Care Quality Indicators in China: A Systematic Literature Review"

_healthcare, 2025, doi:10.3390/healthcare13141637_

Round 1

Reviewer 1 Report

Comments and Suggestions for Authors

Dear Authors,

Thank you for providing me with the opportunity to read this interesting paper. Below, I have listed my comments:

This is a well-written and ambitious systematic review that addresses a timely, underexplored topic: the quality indicators for home-based community eldercare in China. My comments will only focus on the discussion section as I have no point to make about the rest of the sections.

1) While the paper notes the need to localize models, it does not suggest or prototype any specific culturally grounded framework. For instance, how might Confucian ethics, community mutual aid, or filial piety norms be operationalized in a QI system? Theoretical Integration Needs Deeper Cultural Anchoring

2) The reference to OECD indicators is appropriate, but the discussion is superficial; it lacks a detailed analysis of specific domains (e.g., person-centered care, safety) where Chinese practices diverge or align.

3) The lack of caregiver and provider voices is flagged, but no further exploration or proposed strategy (e.g., participatory QI design, stakeholder workshops) is presented.

I hope this feedbakc is helpful.

Author Response

Comments 1:

While the paper notes the need to localize models, it does not suggest or prototype any specific culturally grounded framework. For instance, how might Confucian ethics, community mutual aid, or filial piety norms be operationalized in a QI system? Theoretical Integration Needs Deeper Cultural Anchoring

Response 1:

Thank you for pointing this out. I/We agree with this comment. Therefore, I/we have the correction-page number 17, and line 271-282:

Traditional Chinese cultural concepts such as filial piety, Confucian ethics, and com-munity mutual aid significantly influence eldercare practices in China. For example, filial piety emphasizes children's duty to care for elderly parents, and Confucian ethics stress respect, moral obligation, and social harmony. Operationalizing these values could involve QIs focused explicitly on family engagement (frequency of family visits, satisfaction of elderly individuals with family participation), mutual community sup-port (community-organized social care activities, community involvement in caregiv-ing), and indicators evaluating respectful, dignity-focused interactions between care providers and elderly individuals.

Future indicator development should therefore integrate these culturally grounded elements to create meaningful and relevant evaluation frameworks that genuinely reflect the lived experiences and societal expectations of elderly individuals in China.

Comments 2:

The reference to OECD indicators is appropriate, but the discussion is superficial; it lacks a detailed analysis of specific domains (e.g., person-centered care, safety) where Chinese practices diverge or align.

Response 2:

Thank you for pointing this out. I/We agree with this comment. Therefore, I/we have the correction-page number 19, and line 364-376:

The comparative analysis reveals critical gaps and alignments between Chinese HCEC practices and OECD benchmarks. For instance, in the domain of safety, OECD indicators focus explicitly on standardized tracking of falls, pressure ulcers, and avoidable hospitalizations. Chinese practices show limited and inconsistent adoption of such systematic safety tracking. Regarding person-centered care, OECD emphasizes respect for individual preferences and comprehensive care planning, whereas Chinese indicators primarily address general satisfaction without clear operational measures of personalized care planning. Similarly, OECD effectiveness indicators prioritize sys-tematic tracking of functional status improvement and pain management, areas mini-mally addressed in current Chinese frameworks.

To align Chinese eldercare practices more closely with OECD standards, future QI frameworks should explicitly adopt indicators for systematic safety and health out-come tracking, personalized care planning, and structured care coordina-tion.

Comments 3:

The lack of caregiver and provider voices is flagged, but no further exploration or proposed strategy (e.g., participatory QI design, stakeholder workshops) is presented.

Response 3:

Thank you for pointing this out. I/We agree with this comment. Therefore, I/we have the correction-page number 18, and line 315-332:

Most reviewed studies were geographically limited to specific provinces or municipalities, restricting the generalizability of their findings [35-37]. Furthermore, quality assessments predominantly emphasized the elderly individual's perspective, largely neglecting critical insights from caregivers, family members, healthcare providers, and service administrators. Such limited perspectives risk overlooking the multifaceted and context-sensitive nature of eldercare quality, given that subjective perceptions vary significantly based on personal experiences and health status (Wu, 2018) [38]. Moreover, reliance on predominantly quantitative designs further constrains nuanced interpretations (Tang, 2018) [39]. To address this significant gap, future research should employ participatory quality indicator (QI) design methods. Specifically, stakeholder workshops, Delphi panels, or focus group discussions involving caregivers, healthcare providers, elderly individuals, and administrators can facilitate the co-creation of indicators, ensuring diverse perspectives and practical considerations are effectively integrated. Additionally, qualitative methods such as in-depth inter-views or case studies should be systematically incorporated to capture underlying barriers and challenges encountered by these stakeholders, such as caregiver burden, re-source constraints, and service delivery complexities (Liu, 2020) [14]. Incorporating community-level risk-adjustment factors will further enhance the contextual accuracy and applicability of quality assessments (Guo, 2014) [40].

Reviewer 2 Report

Comments and Suggestions for Authors

Thank you for the opportunity to read your interesting research. While reviewing the paper, I had a few questions and suggestions:

1. Could you clarify the main objective of the study? A clearer statement of the research question would help readers better understand the purpose of the work.

2. In the introduction, you mention an intention to examine quality indicators. However, the results and discussion sections seem to focus more on the model itself rather than on those indicators. Please consider reviewing the overall flow of the paper to ensure consistency with the stated objectives.

3. While the search strategy is described, it would enhance clarity and transparency if it were structured using the PICO framework, with detailed strategies provided for each database or source.

4. Table 1 currently contains a large amount of information, making it difficult to read. It may be clearer to list only the author and year in the table and assign a unique identifier to each study. Additional details, such as study titles, could be moved to an appendix. This identifier could then be used in Table 2 for easier cross-referencing.

5. In the discussion section, you include a table on OECD long-term care quality indicators. This content may be more appropriately placed in the introduction for context, with relevant comparisons and analysis discussed later in the discussion section.

Author Response

Comments 1:

Could you clarify the main objective of the study? A clearer statement of the research question would help readers better understand the purpose of the work.

Response 1:

Thank you for pointing this out. I/We agree with this comment. Therefore, I/we have the correction-page number 3, and line 100-107:

Therefore, the main objective of this study is to systematically identify, synthesize, and critically evaluate the specific quality indicators currently employed in home-based community elderly care in China. This systematic review addresses the central research question:

What specific quality indicators have been used in evaluating home-based com-munity elderly care in China, and how can these indicators inform the development of a comprehensive, culturally relevant, and practically applicable quality assessment framework aligned with international standards?

Comments 2:

In the introduction, you mention an intention to examine quality indicators. However, the results and discussion sections seem to focus more on the model itself rather than on those indicators. Please consider reviewing the overall flow of the paper to ensure consistency with the stated objectives.

Response 2:

Thank you for pointing this out. I/We agree with this comment. Therefore, I/we have the correction-page number 3, and line 108-111:

By explicitly analyzing the indicators themselves—rather than primarily focusing on theoretical models—this review seeks to clarify current evaluation practices, identify critical gaps, and provide evidence-based insights to support future policy and practice enhancements in eldercare quality assessment.

Comments 3:

While the search strategy is described, it would enhance clarity and transparency if it were structured using the PICO framework, with detailed strategies provided for each database or source.

Response 3:

Thank you for pointing this out. I/We agree with this comment. Therefore, I/we have the correction-page number 3, and line 125-131:

The systematic search strategy was clearly structured in March 2025 using the PICO framework to ensure clarity and transparency:

Population (P): Elderly adults aged 60 years and older.

Intervention (I): Quality indicators (QIs) used to evaluate home-based community elderly care.

Comparison (C): Not applicable.

Outcomes (O): Identification, synthesis, and critical evaluation of relevant QIs.

Comments 4:

Table 1 currently contains a large amount of information, making it difficult to read. It may be clearer to list only the author and year in the table and assign a unique identifier to each study. Additional details, such as study titles, could be moved to an appendix. This identifier could then be used in Table 2 for easier cross-referencing.

Response 4:

Thank you for pointing this out. I/We agree with this comment. Therefore, I/we have the correction-page number 6-26, and line 232-440

Comments 5:

In the discussion section, you include a table on OECD long-term care quality indicators. This content may be more appropriately placed in the introduction for context, with relevant comparisons and analysis discussed later in the discussion section.

Response 5:

Thank you for pointing this out. I/We agree with this comment. Therefore, I/we have the correction-page number 2, and line 88-92:

Quality indicators (QIs) play an essential role in evaluating and improving elder-care services. Internationally, organizations such as the Organisation for Economic Co-operation and Development (OECD) have established comprehensive long-term care indicators covering domains like safety, effectiveness, person-centeredness, coor-dination, and equity of care [41-43].